# Diaconal Church Initiatives and Social/Public Welfare in Postwar Japan: A Descriptive Overview

Stéphan Van der Watt [1,2]

1   Department of Practical and Missional Theology, Stellenbosch University, Stellenbosch 7602, South Africa; jsvdwatt@gmail.com
2   Kobe Reformed Theological Seminary, Kobe 651-1306, Japan

**Abstract:** This article reflects on post-WWII developments and the current state of church-related diaconal initiatives in Japan. Pioneering Christians have made significant contributions to the development of social welfare since the Meiji Era (1868–1912). Despite still being a radical minority of around only 1 percent of Japan's population, the nationwide network of Japanese Christian churches, educational institutions, and social welfare organizations makes Christianity's presence felt on a much wider scale. With its focus on postwar efforts, this article gives a brief overview that ranges from education to social reform and medical care, all of which were traditionally incorporated under the notion of "Christian Social Welfare" (*Kirisutokyō Shakai Fukushi*). The research integrates Japanese and English sources in a methodical, rigorous literature study in response to the following main question: Why is there a complicated relationship in postwar Japan between church practices defined as *diakonia* and the work of Christian-based social welfare organizations? This article discovers how *diakonia* as a theological concept is re-orientating the core identity and mission of churches in Japan. A case study from the Reformed Church of Japan's diaconal activities is presented to highlight the conclusion that a complex relationship remains between social welfare organizations and wider church practices enacted under the rubric of *diakonia*.

**Keywords:** postwar Japan; *diakonia*; social welfare; public welfare; holistic care; justice; dignity

## 1. Introduction

Many historical investigations dealing with post-WWII developments related to Christianity in Japan view such developments on a continuum, which evolved from the introduction of the Roman Catholic faith by Francis Xavier in 1549, or with the reintroduction of Christianity under Protestant and Catholic missionaries from 1859. Although these significant eras and their historical continuities indeed need to be understood, an overemphasis thereof inevitably blurs the fact that Christianity in Japan has functioned under a radically changed context since 1945. In addition, Japan's postwar context cannot merely be viewed as an extension of "World Christianity", since the events experienced by Japanese Christians in the aftermath of the war were unique in various ways (Phillips 1981, p. 1). One example that testifies to this truth is the fact that Japan is the only nation in the world whose inhabitants have endured the calamitous reality of two atomic bomb attacks.

This article first provides a brief overview of Japan's socio-economic realities since the aftermath of the two devastating atomic bomb disasters up until today. Second, is a description of the development of postwar initiatives of Christian Social Welfare (*Kirisutokyō Shakai Fukushi*), including such areas as social reform, medical care, and education. The article then proceeds to investigate, in the third and last part, in which ways discourses about *diakonia* and holistic care have theologically re-directed the foundational identity and mission of some church denominations.

In Japan, where natural disasters are very prevalent, diaconal response initiatives have recently grappled afresh with the existential realities of human vulnerability and suffering. The research is situated within the emerging ecumenical discussion of megatrends

related to diaconal practices and the expanding recognition and significance of care under the rubric of *diakonia* (Latvus 2017, p. 11). In this article *diakonia* is understood as a call to action—arising from the gospel of Jesus Christ—which incorporates care for God's whole creation and seeks to combat many forms of injustice and vulnerability. The last part of the article introduces a brief case study of the diaconal activities of the Reformed Church of Japan.

## 2. A Brief Overview of Japan's Socio-Economic Realities after WWII

Japan's recovery from WWII was multifaceted and complex. The nation was both a defeated aggressor and a devastated victim. Suffering, fundamental changes, and preserving Japan's heritage were fused in the aftermath of the atomic bombings and the nation's unconditional surrender.

### 2.1. From Despair to Prosperity within One Generation

Up until 1945, State Shinto(ism) paired the imperial mythology—believing the emperor to have absolute sovereign power—with family ethics based on Confucian values. Japan's domestic religion integrated Chinese religious and European political values with its age-old mythology to perform a dedicated nationalistic purpose. From 1890–1945, all Japanese school children were systematically indoctrinated into this nationalist ethos by the regular, solemn recitation of the Imperial Rescript on Education.

World War II left an indelible destructive impact on Japan, dislocating the lives of millions of people who faced the brutal aftermath of two atomic bomb attacks. The bombing caused lasting wounds but also ended Japan's militarist ambitions to conquer East Asia under its imperial kingdom. Japan had to "embrace defeat", as the title of John Dower's compelling history of its WWII failure, and its subsequent restoration under Allied (mainly American) occupation during 1945–52 (Dower 1999), suggests. The nation was left not only with the enormous job of reconstructing shattered cities and an economy in ruins but also with the healing of its people's torn lives. Somehow the very sense of what it meant to be Japanese had to be renewed (Phillips 1981, p. 3).

A pervasive victim consciousness shot up, as many Japanese viewed themselves as suffering the worst attack in world history. Local poets such as Horiguchi Daigaku (quoted in Dower 1999, p. 119) tried to remedy the pessimistic predicament by inspiring the downtrodden nation to pick their heads from desperate despair (in 1946).

> The country has become small
> and powerless,
> food scarce,
> shame plentiful,
> life fragile.
> Stop grieving!
> Raise your eyes
> to the treetops,
> to the sky.

Dower succinctly summarizes the Japanese people's state of mind as "a populace sick of war, contemptuous of the militarists who had led them to disaster, and all but overwhelmed by the difficulties of their present circumstances in a ruined land. More than anything else, it turned out the losers wished both to forget the past and to transcend it" (Dower 1999, p. 24).

The novelist Ryū Murakami (2002) argues that, during those years of utter hardship directly after the war, no one had anything but hope. Apparently, this hope—to survive and thrive again—pulled many people forward. Over the next couple of decades, following the end of the occupation period (from April 1952 onward), the nation miraculously managed to show remarkable resilience by making an astonishing economic recovery, with

success upon success in terms of its high-paced economic growth in the boom years of its postwar "bubble economy".

This "miracle" happened without the traditional power foundations: no missiles, no raw materials, no geopolitical conquests, no colonies. Indeed—claims Gibney (1985), as well as Mason and Caiger (2011)—humanly speaking, Japan's principal source of energy has been the ingenuity of its extraordinarily talented, disciplined, and highly motivated nation society. However, Korea (South and North) would dispute this claim. Instead, they point to US aid—which Korea (neither South nor, of course, North) did not receive to the same degree.

At any rate, the nation's sustained industrial output and its creativity in new-age consumer electronics received global acclaim by the 1970s. Japan's economic growth sparked a diversity of technological creativity known as *Kaizen*, a notion credited to Toyota Motors, of perpetually improving an existing product. This creative ingenuity has a long history that evolved from Japan's relationship with China, which led to the phrase *Wakon, Kansai* ("Japanese spirit, Chinese learning"). Since the *Meiji* era, when Western scripts coincided with Japan's national rebirth, the archaic trademark was changed to *Wakon, Yōsai* ("Japanese spirit, Western learning").

One outstanding trait during the 1970s and 1980s is that the country achieved a noteworthy balance between a high level of job security for (male) workers and phenomenal economic growth. The state organized its "super stable society" (*chō antei shakai*) around the three pillars of family, corporation, and school. The "salaryman" (*sararīman*) and "education mama" (*kyōiku mama*) were upheld as ideals in this era. Families were reconstructed and progressively lost their traditional educational, economic, socio-political, and caring duties. A new nuclear type of family was formed, which increasingly relied upon external social organizations such as schools and other government institutions. Japan grew into an all-middle-class society, focused on continual improvement in production and sustained long-lasting jobs, social connections, low crime rates, and no war or military engagements. Through its unique "Japanese management style", the country became well known for its vast wealth and its highly advanced consumer culture (Allison 2013).

The impact of the Allied occupation on demilitarization, democratic values and redistribution of power and wealth, paved the way to a thriving, equitable economy and unmatched growth in population health. Bezruchka et al. (2008) deem it as the "legacy of dismantling the prewar hierarchy". Japan's super stable society was aided by a new postwar constitution facilitated by Japanese values. The largely equal spread of wealth has enabled the nation to sidestep much of the socially corrosive effects that income inequality inevitably causes.

Wilkinson and Pickett (2009) have convincingly indicated that the quality of a nation's social relations is built on material foundations because the scale or income differences have a determining effect on how people relate to each other. Wilkinson and Pickett's (2009, pp. 18–24) core finding is that inequality in societies in fact makes a concrete difference to health and social problems. Making this applicable to Japan, in terms of the following indicators (at least up until 2009), Japan was mostly outperforming other top developed OECD nations with regard to the following indicators: level of trust, mental illness (including drug and alcohol addiction), life expectancy and infant mortality, obesity, children's educational performance, teenage births, homicides, and imprisonment rates.

Japan currently boasts a distinguished position among other global leading economic powers. It has the fourth largest proportion of the global domestic product, although it is ranked eleventh in the world in terms of population size (Statista 2021), taking 34th place in the 2022 IMD World Competitiveness Ranking (out of 63 economies) (IMD World Competitiveness Booklet 2022). In addition, socio-politically, the Japanese have gained a constructive benefit from their WWII experience of defeat: most Japanese have consistently made their loss the touchstone for affirming a commitment to peace and democracy. Dower (1999, p. 30) calls this "the great mantra of postwar Japan."

## 2.2. Post-Bubble Challenges and Major Natural Disasters Expose a Fragile Society

However, this bright evaluation does not reflect the full picture. In her book entitled *Precarious Japan* (2013), cultural anthropologist Anne Allison perceptively analyzes some of the contemporary challenges which the nation is facing. Allison notes that postwar Japan was sometimes nicknamed "Japan, Inc." Its amazing productivity was based upon a system of lifelong work attachments. It relied upon the "marriage" between the social factory at home and the postindustrial factory at work. However, this partnership has recently been falling apart, particularly after the economic bubble burst more than three decades ago in 1991. Allison calls this breakup the shift from "Japan, Inc." to "liquefied Japan". Her book's title suggests that "precarity" is a word of the times, worldwide, especially now in Japan. Employment is becoming increasingly uncertain, unpredictable, and risky, although the national unemployment rate has been consistently low over the past decade (currently at less than 3 percent—O'Neill 2022).

Moreover, the *Heisei* era (1989–2019) can be defined by many keywords, a significant one being "insecurity" because of an ever-present threat to safety, specifically induced by natural disasters. There were two major earthquakes—the Great Hanshin-Awaji Earthquake of 1995 that destroyed Kobe City, killed 6434 people, and left thousands homeless, and the Great East Japan Earthquake of 2011 (also called "3.11" because it occurred on March 11), which caused an enormous tsunami, a loss of almost 19,000 lives, and a disastrous nuclear crisis. In addition, the worst act of terrorism in modern Japanese history occurred in 1995—in the form of a sarin gas attack — by the doomsday cult *Aum Shinrikyō* on the Tokyo subway system (Martin 2019).

The 1990s also introduced Japan's era of unsafe nationalism, its new era in Japanese-style management, and the labor "big bang". The so-called liquefaction or flexibilization of work and life ensued. With growing unemployment, near-zero interest rates on investments, and many pension funds close to bankruptcy, the postwar generation began to question why they had worked so hard to achieve so little. According to Takeda (2008) more risk-taking, individual responsibility, and entrepreneurship were expected to rid the Japanese of the old "dependency culture". These aforementioned factors contributed to why this decade is also called the "lost decade" (*ushinawareta jyū nen)*: a period of economic stagnation after the economic bubble burst. Significantly, Japan's lost decade is now being compared to the current socio-economic crisis in the United Kingdom (Rees et al. 2023).

A long period of deflation ensued, with Japan's public debt-to-GDP ratio hovering far above 100 percent for more than 26 years up until today (IMF 2023). This situation also led to a more disparate society (*kakusa shakai*) because of widening income inequality and a shrinking population, consisting of a dangerously high proportion of elderly people. Recently the birth rate dropped to 1.34 (children per household) in 2021, far below the required 2.1 for a stable population and the lowest on record since 1899 (Yamamoto 2022). This drop prompted Prime Minister Fumio Kishida to take drastic measures of "unprecedented levels" (Otake 2023), asserting that it is "now or never" for this gerontic society (Murakami 2023). With almost 30 percent of people older than 65, O'Neill's (2023) contention seems to be spot on: "with high age comes less capacity, and Japan's future enemy might not be an early death, but rather a struggling social network".

## 2.3. Current Realities and Challenges

The above-mentioned compounding issues elicit far-reaching anxiety because of the realities and fear of an increasingly unequal society. In 2017, the richest 20 percent of the Japanese population earned more than six times as much as the poorest 20 percent (see Chiavacci and Hommerich 2017). Social scientists generally agree that inequality has increased, but to what extent has been a controversial point (2017, p. 3). However, there remains no doubt that the mood during the first decade of the new millennium was distinctly different from the first postwar years. Murakami (2002) asserts that Japanese people have become consumed by materialism (since the 1980s) to such an extent that the urge and hope for anything beyond private acquisition were fading away. But how does this con-

sumerism influence people's emotional and spiritual lives? Today, it seems that hope is the only thing Japanese people do *not* have, according to Murakami. Has the emphasis on accumulation made the existential need to hope "unnecessary" for many Japanese?

Japan's efficient global market capitalism system has been thriving on instant gratification and technology, but that hype did not last. The fragility of Japan as a superpower surfaced gradually. The national television network (NHK) broadcast a documentary in 2006 which coined the phrase "working poor", following an OECD report which stated that the rate of relative poverty in Japan was among the highest of its member nations. Another new phrase entered the Japanese lexicon: "net cafe refugee" (*netto kafē nanmin)*, representing a multitude of laid-off workers, the drifting poor, who made Internet cafes their transitory places of refuge and otherwise would have become homeless (*Heisei hangover*).

Most of the above-mentioned factors have caused a sense of being out of place for many Japanese, a kind of disconnectedness (*ibasho ga nai*) that deepens the pain of social isolation and loneliness. Another NHK special series in 2009 even labeled Japanese society as relationally disengaged (*muen shakai*). In the strikingly competitive, education-credentialist Japanese society, stress levels are high. This situation leads to overwork (for adults) and extra class (for children) culture, which causes intimacy in family relations to dwindle. Deep company loyalty paired with disproportionate working hours and overtime expectations breeds a culture in which there is even a term for sudden death by overwork (*karōshi*). A white paper released in 2022, reports a larger than 60 percent increase in the number of people liable for mental illness benefits induced by work-related stress, for the decade between 2009–2019 (The Japan Times 2022).

Loneliness also surfaces through prevalent phenomena such as *hikikomori*, defined as "adolescence without end" in the subtitle of his book by popular psychiatrist Tamaki Saitō (Saitō and Angles 2013). This label refers to socially withdrawn individuals who refuse to leave their homes or bedrooms for anywhere from six months to many years. These young people are also called the "homeless inside of home" numbering between 500,000 and 2 million (Ismail 2020; Borovoy 2008). Loneliness is obviously experienced by many more Japanese than just socially withdrawn youth. The impact of the Corona pandemic and multiple other factors have also brought the intense socio-economic impact of loneliness to the fore in many other contexts, e.g., in Europe (European Commission 2023).

In Japan some researchers—such as anthropologist Chikako Ozawa-de Silva (2021)—have even labeled it a loneliness epidemic, arguing that Japan can be viewed as a society whose people lack care (by society as a whole) and whose "structures promote a sense of loneliness rather than one of belonging and connection". Ozawa-de Silva further contends the 3.11 disaster and its concomitant displacement, as well as the "perceived mishandling of the situation by government, media, and corporations, led to feelings of isolation, abandonment, and hopelessness" (2021, pp. 6–7). She connects this collateral sense of disorientation and despair to the prevalent difficulty of living (*ikizurasa*) and lack of life meaning (*ikiru imi*) that lead many Japanese to social disconnection and even suicide.

Instances of people dying alone—from abandonment, lack of care, or simple "disinterest"—as well as the fear of solitary death (*kodokushi*) are rising (Allison 2013, p. 40; Ozawa-de Silva 2021, p. 136). In general, social ties between people get superficial because of being overly busy. Family life has obviously not been left unchanged either. The "salaryman" and "education mama" morphed to make space for *ikumen* (men who take care of their children) because of the drastic drop in households with male breadwinners and non-working female spouses (between 1990–2017). Concurrently, there has been a steady increase in dual-income households (Kawasaki 2019).

Japan had the largest gender gap among advanced economies in 2022, ranked 116th among 146 countries (WEF 2022, p. 10; The Japan Times 2023). The prolonged coronavirus pandemic is likely to lead to a widening of this gender gap. In addition, the COVID-19 pandemic has driven (already high) suicide rates up, especially among women and youth (Yoshioka et al. 2022). Not surprisingly, critical voices from outside have recently been

raised, and the pressure to adapt to a globalizing world—as a country with a resolute homogeneity (98.5 percent is ethnically Japanese)—has been stiff. For example, Noah Sneider (2021), *The Economist*'s Tokyo Bureau Chief, asserts that Japan's cultural essentialism is for Japanese "both a source of pride and a cover to ignore examples from outside . . . Japan's treatment of women is retrograde, its protection of minority rights weak, its government services archaic and its climate policy dirty. Many institutional frameworks are stuck in the past." Therefore, Sneider echoes Yoshimi Shunya of the University of Tokyo who contends that the current *Reiwa* era (since 1 May 2019)—characterized by its post-growth economy—needs to evolve from the faster, higher, stronger motto of the Shōwa period (1926–1989) to "diversity, resilience and sustainability".

Allison (2013, p. 59) even calls Japan "post-welfare, post-family and post-relational". The following question comes to mind: how can the church be(come) a place (*ibasho*) of meaningful connectedness, where lonely people gain social recognition, belonging, and hopeful support in hard times?

### 3. Postwar Initiatives of Christian Social Welfare (*Kirisutokyō Shakai Fukushi*)

*3.1. The Nature of Religiosity in Japan*

The diversity and complexity of religion in Japan are not easy to grasp. There is no attempt here to provide an extensive overview of Japan's religious landscape, since, as Mullins et al. (1993, p. vii) rightly affirm, "the array of religions and religious phenomena in Japanese society is staggering in its richness and variety", therefore its explication far exceeds the scope of this article (see, e.g., Inoue 1990). Suffice it to say that religiosity in Japan comprises a blended variety of at least five main streams, namely ancient folk religion, Confucianism, Buddhism, Daoism, and Shinto (meaning the way of the *kami* [gods]) (Earhart 2014).

No endemic notion of "religion" (*shūkyō*) was even in use before the *Meiji* era. After that time (since around 1868 onwards) the three ancient doctrines of Confucianism, Buddhism, and Shinto—which uniquely existed without interruption throughout Japanese history—were transformed: Confucianism became a philosophy, Buddhism developed into a religion, and Shinto was divided into secular political and religious systems (Josephson 2012, p. 258). In contemporary Japan, these religious traditions are fused with indigenous practices in a huge smorgasbord of traditional religious institutions and new religious movements (see, e.g., Inoue 1990).

Although Japan is often deemed a very homogeneous society, the ways of being religious are extremely varied and reflect both continuity and discontinuity with the past. Common religious rituals are typically viewed and performed as cultural customs, and they are inextricably intertwined with an apparent self-identified secularity (Mullins 1994, p. 65). Ironically, in government surveys most Japanese claim to be "irreligious" and the nation is often viewed as the most secular of all modern societies, leading some researchers to call it the most religious atheist country in the world (Coslett 2015). However, understanding "secularization" in the Japanese context involves different discourses from those typical to the West. The postwar situation accelerated secularization by the removal of State Shintō during the Allied Occupation (1945–1952) and freed the Japanese from being bound to restricting religious affiliations offered by public institutions (see, e.g., Fitzgerald 2003).

Within this mixture, Christianity represents a tiny minority of about 1 percent of the population and churches clash with the pervasive presence of a legion of superstitions, myths, and syncretistic cults that threaten their survival by thwarting gospel truth (see Shimazono 2004). In the process of secularization, as Sherrill (2003) observes (in Mullins 2003, p. 167) "the pursuit of material goods captured the interest of society, the church and its evangelistic campaigns lost their appeal". Recent church statistics seem to echo this sentiment. Comparing the number of believers in the United Church of Christ in Japan (UCCJ, *Nihon Kirisuto Kyōdan*), the biggest Protestant denomination, there was a 25 percent drop

between 1995 and 2018. Concomitantly, the number of congregants older than 70 years of age has increased from 31.4 percent to 45.8 percent (UCCJ 2021).

The focus of what follows in this article is to understand one aspect of such marginalized Christianity's scope and influence in Japan, namely social welfare.

*3.2. Primary Areas of Social Welfare Involvement Pre-WWI: Social Reform, Education, and More*

The notorious "Black Ships" (*kurofune raikō*) of Commodore Matthew Perry of the US Navy forged open diplomatic ties and trade agreements with the government of Japan (Tokugawa Shogunate) in *Edo* Bay during 1853–1854. This happened after almost two and a half centuries of self-isolation (*sakoku*), which included severe persecution of virtually the whole Christian population (Mason and Caiger 2011). The Christian dilemma in Japan, however, did not end there. Mullins (1994, p. 66) explains that "Although the edict prohibiting Christianity was rescinded by the government in 1873, Christianity continued to be popularly understood and referred to as a heretical and evil religion (*jakyō*). Protestant missionaries, as the most recent 'carriers' of a deviant religion, faced a difficult task".

What followed thereafter was the *Meiji* era, during which a plethora of mission organizations from numerous countries initiated evangelistic and missional involvements in Japan. Such missionaries and their partners not only planted many local churches but also started countless Christian schools, universities, medical facilities, and other social welfare institutions. By the first half of the twentieth century, many of these institutions had become self-supporting.

Traditionally, social work was performed on a family or neighborhood basis. But Christians took the "radical" initiative to care for persons outside of one's clan or kinship circle. Indeed, various Christian churches have been involved in the needs of suffering people long before the emergence of social welfare professionals. During the *Meiji* era, government-formulated social welfare policies did not exist. It was up to pioneering individuals and groups to initiate private social welfare organizations. In fact, some of the first programs for the reformation of prisoners (by Caroline Macdonald, 1874–1931, who also started the YWCA in Japan) and the first orphanages (by Jūji and Fudeko Ishii, father of child welfare, mother of mentally handicapped children at Okayama orphanage) were started by Christians. Furthermore, during 1880–1930, pioneers such as Takeo Iwahashi (founder of the Light House for the Blind in Osaka), Kōsuke Tomeoka (father of rehabilitation of juvenile delinquents), Gunpei Yamamuro (founder and developer of the Salvation Army in Japan), and Toyohiko Kagawa (evangelist, social and labor reformer, peace advocate) did enormous work that was also recognized globally (Phillips 1981, pp. 7, 83; see also Hastings 2013).

During the ensuing decades, government responsibility for welfare increased, which caused these above-mentioned grassroots efforts to decline. In this regard, the passing of the Poor Relief Law (*kyūgohō*) in 1929 and the authorization of a series of welfare laws (*fukushi sanpō*) between 1947–50 were especially significant. Notwithstanding, some Japanese scholars specializing in Christian social welfare research, such as Hiroaki Sugiyama (2015), argue that these laws and welfare policies were not able to reach all people who needed help. He, therefore, highlights the ministries of less well-known Christians, whose work in rural areas especially is often overlooked by researchers. Sugiyama's focus on social welfare via "farm village evangelism" (*nōson dendō*) from the 1930s until the 1950s deserves special mention as a reminder that the influence of Christian mission through social welfare lies deeper than what is mostly known only superficially (because of its visibility in metropolitan areas or via well-known institutions).

What is evident in the sketch above is that elaborate ministries of Christians contributed widely to social welfare in Japan since at least the 1870s. This Christian contribution to social welfare in Japan is widely recognized and has been integrated since long before WWII, as a necessary and fundamental part of Protestant and Roman Catholic mission in Japan, with participation from a very wide range of mainstream, independent, charismatic, Pentecostal and other denominations and organizations (see Tashiro 1989 for

a broader historical overview). Undeniably, at least until the 1930s, "despite its size, an impressive nationwide network of Christian churches, schools, universities, and organizations for social welfare gave Christianity a visible presence in Japanese cities that belied its few believers" (Ion 2003, in Mullins 2003, p. 71). The following section takes a closer look at postwar developments that influenced the variegated range of work conducted under the notion of "Christian Social Welfare" (*Kirisutokyō Shakai Fukushi*).

*3.3. Initial Postwar Christian Social Welfare Developments*

3.3.1. Occupation Era (1945–52)

As mentioned in the introduction, the post-WWII climate concerning religion—Christianity in particular—and its influence in Japan changed dramatically. After the war, new policies of religious freedom greatly influenced the nature and content of Christian-based institutions' work and ethos. Hastings and Mullins (2006, p. 18) report that more than 1500 new (Protestant) missionaries were sent to Japan during 1949–53. These missionaries' work was concentrated in major city areas, and the demographic shifts accompanying the postwar restoration created a favorable situation for missionary activities (Mullins 1994, p. 71). However, a constructively critical and balanced view of such missionaries' work is needed. That is why Mullins (2003, p. 145) aptly warns elsewhere, "Without denying that Protestant missionaries have also been 'transformers of culture' through their activities in the fields of education (particularly for women) and social welfare, the fact remains that their understanding of the relationship between the Gospel and Japanese culture has been fundamentally negative. Missionary theology and practice have tended to emphasize a total discontinuity between the Christian faith and Japanese religious traditions and practices". This shadow side of Protestant missionary activities is worth exploring in detail, but space in this article is too limited to do so (see Mullins 2003, pp. 143–62).

Shortly after the end of WWII, Occupation forces—led by its SCAP (Supreme Commander, Allied Powers)—enforced new legislation that assured the separation of government control from all religious organizations. These new laws were detailed in the Japanese constitution and implemented via the Religious Corporations Ordinance (in 1945), later reinstated by the Religious Juridical Persons Law (in 1951). Although the SCAP Religions Division's work was based on this strict separation of politics and religion, its head General Douglas MacArthur often advanced Christianity in Japan by calling for "thousands of missionaries to transform the heart and soul of Japan" (Woodard 1972). The pervasive self-scrutiny initiated by the crushing defeat and the emperor's renunciation of his divinity (1946), led by MacArthur, launched a religious reawakening, including a "'Christian boom' of considerable proportions" (Phillips 1981, pp. 4–5).

What ensued was not merely the relief and reconstruction of Christian buildings but also a dynamic expansion of Christian literature, theology, lay witness, art, and social service. The Ministry of Education's "Order No.12", which prohibited the teaching of religion or courses in religious education taught since 1899 in all public and private schools, was abolished in November 1945. Along with the postwar baby boom and gradual economic recovery, an education explosion occurred, with Christian schools and universities among the main recipients of resources (see Phillips 1981, pp. 50–80 for an overview of this spectacular expansion which he dubs an "exceedingly complicated story"; furthermore, see Mullins (1994) for an excellent exposition of the problem of secularization of Christian schools after this initial boom in Christian education).

In a similar way, as was the case with education, social welfare was another area that benefited from SCAP policies that promoted society's welfare for the sake of "democracy". When faced with the unsurmountable task of rebuilding a destroyed postwar Japanese society, many Christians were already acknowledged leaders in fields of social welfare work. New legislation under the Livelihood Protection Law (*Seikatsu Hogo Seido*) (1946) and other welfare laws made it possible for many new Christian-based social work agencies to be started during the Occupation era. The proportion of Christians among social

work staff was high, "for such humble work for long hours at low pay required a personal commitment that Christian faith helped to supply", according to Phillips (1981, p. 7).

In the early postwar years, with many suffering and displaced people in ruined cities with decimated infrastructure, conducting social welfare work (alongside essential relief efforts) was inescapable. An added problem was that traditional ways in Japanese society for providing such services were no longer sufficient. Two such methods were (1) the ancient pattern of giving responsibility for people with social needs to that person's next of kin (based on the Seventeen-Article Constitution in 604 AD, outlined by Prince Shōtoku, 573–621) and (2) philanthropy through donations of charitable organizations, mostly Buddhist or Christian.

However, the scale of the war devastation was so sweeping that these two methods were totally overwhelmed. For this reason, the establishment of a totally new foundation for social work in Japan evolved, namely that of government responsibility for the nation's welfare. However, Christian-based private social welfare institutions were not thereby ignored. In fact, Christians kept serving on advisory committees for the Ministry of Welfare, and Christian institutions kept on making a constructive contribution, albeit under very demanding circumstances. Many such institutions had to be restarted after the war (Phillips 1981, pp. 82–83).

The Occupation era eventually witnessed the establishment of countless private voluntary Christian Social Welfare agencies, which partnered with the efforts of public authorities to address Japanese society's staggering needs at hand. Such organizations included hospitals, special homes for people challenged by physical or mental disability, orphanages, and social centers. Many of these institutions were grouped together in the ensuing decades under systems such as Japan Church World Service (later known as the Division of Christian Service of the National Christian Council of Japan). There are countless other groups that were involved too, including Licensed Agencies for Relief in Asia (LARA), Lutheran World Relief, and Caritas Japan (see Abe and Okamoto 2014, pp. 203–54). Although the legal status of such voluntary agencies was at first uncertain for many years, it was later dealt with by the establishment of "social welfare juridical persons" (*shakai fukushi hōjin*).

As Phillips (1981, pp. 86–87) points out, Japan's new welfare system was based not on private philanthropy or next of kin relations, but on the principle of social justice:

> Adequate standards of livelihood were supposed to be established by citizens' rights, not by donors' goodwill. For Christian social work agencies, this meant, for instance, that their activities could no longer be understood as dispensing charity or as a means for promoting conversions. Christians would continue to offer their services on the basis of love and compassion, but with the understanding that the recipients were entitled to these services on the basis of their legal rights. The humblest of God's, children, must be served, but so also must Caesar. The new principles of social work would cause Christians to re-examine the theological bases on which they had been conducting social work as well as the methods they had been using.

Indeed, that was and is the case ever since, namely that Christians who are involved in Christian Social Welfare agencies in Japan have been oscillating in the tension field between state and church expectations and obligations.

### 3.3.2. Period of Rapid Economic Growth Exposes the Need for Social Welfare in the Post-Occupation Era (1952 Onwards)

The end of the Occupation period led to a new phase, in which the outbreak of the Korean War (June 1950) shifted the SCAP's relationship with Japan in the direction of allies instead of foes. The post-Occupation era also brought more freedom to Christian organizations to expand in their own way, alongside the brimming urbanization and potent economic developments that made it possible for more people to support the work of churches and voluntary associations. In the first decade after the Occupation, after the new legisla-

tion for the social welfare system took root, Christian social work agencies established a new legal basis for their work whereafter they expanded their facilities for the great needs before them.

Kumazawa and Swain (1991, pp. 191–226) compiled a very helpful, albeit partial summary of some of such welfare services, ranging from work with people living with disabilities or with day laborers, to hospices and terminal care, as well as professional telephone counseling (*Inochi no Denwa*). Included in this list are three organizations that started in partnership with the Reformed Church in Japan—namely Seikeikai Social Welfare Institution, Yodogawa Christian Hospital, and Shizuoka Centre for the Blind—one of which will be explored further later in the article, as part of a brief case study. A more detailed overview of the spectrum of Christian Social Welfare is not possible here, but the Christian Yearbook (2022) gives an in-depth description thereof (in Japanese).

In later years after the Occupation, Japan's economic boom developed at a soaring speed, as was explicated in the first part of this article. Events such as the 1964 Tokyo Olympic Games and the 1970 Work Expo in Osaka shone as symbols of Japan's magnificent return as a full member of and contributor to international goodwill. However, pockets of poverty among coal mine workers, fishermen, and farmers (among others) also lingered in the shadows of economic expansion.

Moreover, in the hub of its major cities remained the predicament of thousands of day laborers, namely in Kamagasaki (Osaka), Sanya (Tokyo), Kotobuki (Yokohama), and Sasajima (Nagoya), who constructed these great world exhibitions of 1964 and 1970. In these gathering places (*yoseba*) the system of day labor work became the lifeline of many men who suffered from the uncertainties and physical toil of construction work (Chamberlain 1994). After various riots broke out in Sanya (in 1960), the *yoseba* came under the spotlight of government attention and wider public awareness. Much of the welfare work across the above four areas across Japan focused on problems of discrimination and human rights as well as the education, health, and counseling of day laborers (Koyanagi 1991, pp. 192–303).

*3.4. Influential Academic Societies Directing Postwar Christian Social/Public Welfare Developments*

3.4.1. The Japanese Society of Christian Social Welfare (JACSW)

In general, most Christian social workers operated within the national social welfare system which gave them access to government aid and services. But this systemic affiliation also made these Christian workers liable to adhere to governmental supervision and control. During this time, the Japanese Society of Christian Social Welfare (JACSW) was established (1960), to support the general development of social welfare. JACSW's further aim was to promote scientific research and practice in social welfare, based on the Christian gospel. This organization (still) annually publishes its "Research in Christian Social Work" (*Kirisutokyō Shakai Fukushigaku Kenkyū*), an important resource for the study of Christian social work in Japan.

One of JACSW's founding members was Shirō Abe, who from 1957 served as director of the Yokosuka Christian Community Center (which was started by Methodist missionary Everett Thompson in 1948). At this center, Abe developed a pioneering program of integrated community services while he was developing the philosophy behind it. Abe also served on the World Council of Churches' (WCC) Central Committee as a lay member and learned much about similar work globally. More recently (in 2014) he co-edited a very significant publication entitled, *History of Japanese Christian Social Welfare* [*Nihon Kirisutokyō Shakai Fukushi no Rekishi*]. This book includes many contributions from influential JACSW members. In the introduction Abe and Okamoto (2014, pp. 1–2) summarize JACSW's main members' shared convictions—as they celebrate and reflect upon the organization's 50th anniversary—as follows:

> In the welfare state system: (1) the situation was beset by deep-rooted ideas that welfare was the responsibility of the state and that Christianity should hand over

its projects to the government and concentrate on evangelization; (2) social services that did not evangelize or cooperate in proselytizing questioned the theological basis for their "Christianity"; (3) problems were raised by non-Christian researchers who argued that religious education in institutions violated the Constitution. Furthermore, in my case, (4) in a local community I was faced with the difficult problem of adapting Christianity to the local culture and struggled to solve it. (my translation)

These last words of Abe echo the sentiment of Endo Shūsaku (1923–1998) who, in his globally renowned book *Silence* (1966), highlights the core conundrum of transplanting the "sapling" of Western Christianity into the "swamp" of Japan. More specifically, Abe's summary illuminates some of the huge challenges that Christian Social Welfare organizations face in a Japanese context where Christians represent a radical minority in society.

In a similar vein, Abe's colleague and co-editor Eiichi Okamoto (Abe and Okamoto 2014, pp. v–vi) writes as follows about the uniqueness of Christian Social Welfare practice in Japan:

It can be said that many pioneers not merely remained spectators of the welfare challenges of their time but chose to practice welfare in answer to Christ's call, as a sign of their faith. Christian social welfare practice has two dimensions: love and justice. The former is conceptualized today as "neighbourly love," as indicated in the Gospel of Mark, chapter 10, where it is written: "Not to be served, but to serve..." In Greek, the word "serve" is rendered "*diakonia*". *Diakonia* is an action of love towards the small and weak. In this (action) justice is concealed . . . . (my translation)

Okamoto relates welfare practices to biblical examples such as Jesus's washing of his disciples' feet and the parables of the Good Samaritan and of the lost sheep, each of which concretely exemplifies what Christian service means. He views this Christian motivation behind welfare practices as the root of volunteerism, individually or as an organization, beyond any ethnic or other boundaries. More significantly for this article, Okamoto directly brings Christian Social Welfare practice in Japan into conversation with the concept of *diakonia*.

Abe (1989) highlights that social welfare initiatives should not be focused merely on alleviating economic poverty but also on the deeper existential needs of human existence, i.e., the "poverty of the heart/soul" (1989, p. 137). Therefore Abe, too, is convinced that the diaconal church is called to connect social welfare and Christianity as a serving community that embodies the love and compassion of Christ, whilst keeping a critical distance towards societal systems and structures.

From the arguments above, it becomes clear that leaders in Christian Social Welfare, such as Abe and Okamoto, have been continuously grappling with the fundamental dilemma of finding a constructive balance between serving Christ (Head of the Church) and serving the Emperor (representing the State) in Japan.

### 3.4.2. Centre for the Study of Public Welfare

The year 2000 brought the initiation of the "Basic Structural Reform of Social Welfare" (*Shakai Fukushi Kiso Kōzō Kaikaku*). New philosophies of decentralization, pluralism, and internationalization sought to direct social welfare into the twenty-first century, including new forms of community-based integrated care. The perspective of "Public Welfare" as a philosophy necessary to integrate citizen-oriented social welfare strongly came to the fore (Inoue 2011).

In line with these developments—that represent a major turning point not only for social welfare but also for Japanese society as a whole—the Centre for the Study of Public Welfare was founded by Prof. Hisakazu Inagaki in 2010, at Tokyo Christian University (TCU). Inagaki is a church member of the Reformed Church in Japan and has sought to make his academic work relevant in some ways to the broader Church. The aim of the

Centre for the Study of Public Welfare is to provide philosophy and practical knowledge for activities based on Public Welfare Studies. The Centre also contributes to the development of human resources (e.g., NPOs, private companies, researchers). TCU is one of a handful of tertiary institutions that attempt to bring Public/Social Welfare Studies into conversation with church-related *diakonia*. One other example in Kwansei University's *diakonia* program.

Institutionally, the "Basic Structural Reform of Social Welfare" caused a shift away from welfare based on "measures" by the state towards welfare based on free "contracts" between users and service providers. Fundamentally, it was a shift from a public (government) dominated society to a society in which the citizens themselves took responsibility for their own welfare. However, due to their long-standing dependence on public services, the Japanese lack the awareness and initiative to create a system of self-government in which citizens themselves cooperate and help each other—a necessary component of a citizenry-driven society. There is a general shortage of workers in the field of welfare, a lack of financial resources, and an increase in the number of isolated and vulnerable people. In this context, while community-based NPO/NGO activities and a younger generation willing to take an active public role are hopeful for the future, such activities have yet to become widespread enough (Inagaki 2010a).

According to Inagaki, the power of religion, which in other parts of the world generates fraternity and solidarity in society, is weak in Japan, creating a huge void in morality and the spirituality that sustains it. This religious weakness is why Inagaki (2012, pp. 3–7) challenges churches (including the RCJ) today to re-envisage their relation to public welfare. He bemoans the fact that churches in Japan struggle to integrate deeds (acts of loving service) with their words (gospel witness) in a holistic way. Inagaki's vision includes a shift away from the individual to public faith in action. He holds Toyohiko Kagawa up as a model of such public faith. Every Christian believer cannot do what the charismatic revolutionary Kagawa did. However, relating Inagaki's views to, for instance, the field of holistic apologetics, fresh potential can possibly be unleashed through *diakonia*. We know Christian witnesses should be present in word *and* deed. When, for example, after 3.11 many Christians showed solidarity with disaster victims in the Tōhoku region, holistic apologetics were embodied. Shoichi Konda (2017, p. 7) aptly contends that these Christians' ministry of empathetic presence showed something of their holistic understanding of the gospel—which in turn is instigating a transformation in basic church structures.

On another level, Inagaki concurs with Shirō Abe's views that *diakonia* is the key theological framework to churches' public and/or social welfare involvement. Inagaki (2012, pp. 159–64) refers to the parable of the Good Samaritan when he explains the need for *diakonia.* He focuses on the role of the innkeeper (Luke 10:35) as a caregiver in the story, and moreover, he interprets Jesus's command to the expert in the law (verse 37)—"Go and do likewise"—as the command to care. This command, argues Inagaki, is closely linked to Christ's Great Commandment to evangelize, therefore *diakonia* (loving service) is viewed as integral to God's mission in this world through us humans. *Diakonia*, as the Christian ethic of care, is displayed at its best via the field of welfare, asserts Inagaki. The Church is called to provide a moral compass at the core of society via an ethic of care. Henceforth, Christians in Japan need not fall into the trap of getting paralyzed or locked in by a "minority complex" but are rather encouraged to participate in community welfare and care, creatively and constructively (Inagaki 2012, p. 165).

Although Inagaki's argument is very relevant in terms of the Japanese context, particularly concerning the relation between social/public welfare and *diakonia*, other research which has been conducted since John Collins's groundbreaking assertions in 1990 clearly indicates that *diakonia* cannot be confined to one core meaning, such as caring service, which is the only dimension to which Inagaki refers. Collins' research is based on an elaborate analysis of related *diak*-words (*diakonos, diakonia,* and *diakonein*) that existed in Greek literature between 400 BC and 400 AD. Collins initiated a fresh interpretation focusing on the work of a messenger. A servant (*diakonos*) fulfills the role of a "go-between" that is authorized by God, in whose service he/she stands.

Collins' research thus challenges the idea that *diakonia* exclusively refers to humble service (social-caritative or caring activity) that is only based on a traditional interpretation of Acts 6. He particularly objects to the direct identification of contemporary social-caritative deacons with the early church's *diakonos*. It is now widely accepted and emphasized that the notion of *diakonia* can refer to different ministerial or administrative roles, performed by emissaries or intermediates, based on Collins' re-interpretation (see, for example, Nordstokke 2016, pp. 147–48).

However, the exact exegetical understanding of the *diak*-word group remains complex and a further discussion of these important developments is beyond the limits of this article (see Collins 1990). Returning briefly to Inagaki's creative initiative in terms of Public Welfare, suffice it to say that his assertion—that *diakonia* is the Christian ethic of care, embodied in various ways in the field of welfare—grasps the core. But, simultaneously, the meaning of *diakonia* can and should be expanded to include a wider range of meanings and practices of diaconal care.

*3.5. Further Reflections*

This article stated at its outset that a core question it attempts to answer is: why does a complicated relationship exist between church practices defined as *diakonia*, and the work of Christian-based social welfare organizations in postwar Japan? In the following Section 4, this attempt will ensue. Although the momentous social welfare work of protestant individuals promoted charitable works and founded many influential agencies, in reality, such activities did not always "trickle down" into the theological formulation of the doctrine of the Church. This reality caused tension about what Christian social welfare, which later also became known as *diakonia*, means in practice. One likely reason that caused or worsened this complex relationship of creative tension is that many Japanese church leaders and theologians, especially during the first postwar decades, did not view diaconal or social welfare actions as a central part of the Church's calling in the world.

For instance, one authoritative Japanese systematic theologian, Yoshitaka Kumano (1899–1981), argued that, although Christian voluntary groups were to perform social work activities, such work is not part of the Church's mission program and hence not part of the essence of the Church (Kumano 1960). This view, i.e., that diaconal or social welfare activities do *not* constitute the core of the Church's identity and existence, obviously does *not* reflect the full range of theological perspectives or arguments on this topic, neither in the context of Japan nor globally. This gap—between views like Kumano's and others that represent converse interpretations—will now be addressed.

**4. Holistic Care and Church-Based Diaconal Responses in Japan and Beyond**

*4.1. Ecumenical Trends in Diakonia*

The 1980s represent a turning point in the ecumenical understanding of *diakonia*. Kjell Nordstokke (2016, p. 145) believes that a paradigm shift took place during that decade in conceptualizing global *diakonia*, with emphasis on three basic dimensions:

1.  Its ecclesial dimension: the Church is by her very nature diaconal.
2.  Its holistic dimension: *diakonia* integrates the Church's whole mission in the world.
3.  Its prophetic dimension: diaconal actions oppose injustice and the abuse of power, and boldly defend the cause of suffering and marginalized people.

From the 1980s, the ecumenical movement—primarily the WCC and the LWF (Lutheran World Federation—provided opportunities for discussing these three dimensions. Influenced by Orthodox theology, the message from the Vancouver Assembly in 1983 defined *diakonia* as follows:

> The "liturgy after the liturgy" is *diakonia*. Diakonia, as the church's ministry of sharing, healing and reconciliation is of the very nature of the Church. It demands of individuals and churches a giving which comes not out of what they have, but what they are. Diakonia constantly has to challenge the frozen, static,

self-centered structures of the Church and transform them into living instruments of the sharing and healing ministry of the Church. Diakonia cannot be confined within the institutional framework. It should transcend the established structures and boundaries of the institutional Church and become the sharing and healing action of the Holy Spirit through the community of God's people in and for the world. (Gill 1983, p. 62)

This statement clearly expresses the ecclesial (i.e., church-based) and missional character of *diakonia*, with mission understood as the action of the Triune God, and not primarily as an activity initiated by the Church. Today, this conviction—that *diakonia* is at the heart of the Church but simultaneously, through God's common grace, reaches far beyond the Church's institutional frameworks—is shared broadly by theologians and church organizations. The missiological and ecclesiological dimensions of *diakonia* were also re-affirmed strongly by the WCC at its latest two consultations on *diakonia*—held in Sri Lanka in 2012 and in Germany in 2022—resulting in its latest publication on this topic titled *Called to Transformation: Ecumenical Diakonia* (WCC 2022). As a gospel-centered appeal to action, diaconal initiatives respond to challenges of human suffering, vulnerability, and injustice, including care for all of creation in a disaster-prone Japanese context.

*4.2. The Need for Diaconal Care in Japan Today: Re-Directing the Foundational Identity and Mission of the Reformed Church in Japan (RCJ)*

This article's Section 2 explained how Japan is facing a wide-ranging array of socio-economic challenges that touch the very fiber of Japanese people's everyday lives. The need for care in multiple ways is self-evident. Psycho-spiritual needs have surfaced more poignantly since the Kobe-Awaji-Hanshin Great Earthquake disaster (1995), after which the concept and practice of heart care (*Kokoro no Kea*) has evolved in Japan. Since then, the term "care" (ケア in Japanese katakana, the writing system for foreign loanwords) has been affixed to many other areas of daily life, including medical (医療ケア), psychological (心理ケア), and social/public welfare (社会・公共福祉ケア). Inagaki (2010b, pp. 86–87) identifies the need for more detailed studies on the variegated notion of care to better understand the concept in Japanese society.

One indication of this new awareness concerning care, within a church context, is the 70th Anniversary Declaration of the Reformed Church in Japan (RCJ), which was published in 2016 in the wake of the catastrophic 3.11 triple disasters. The Declaration includes a specific mention of *diakonia* for the first time in RCJ history, describing the work of serving with love (*diakonia*) as the essence of a church living out (of) the gospel (RCJ 2016, pp. 135, 176–77). Faith communities are learning anew how to be relevant within a post-3.11, post-Corona society. Up until now (since its founding in 1946) the RCJ has created a very stable foundation. Teachings in fields of dogmatics, confessional creeds, and church history have been strong emphases. This focus was necessary and stays vital in such a context where Christianity has a small demographic presence, lacks socio-political power, and needs to survive strong onslaughts from other religions.

But a new era has dawned and the RCJ faces new challenges. Those challenges include becoming more intimately involved in the existential realities in society and grappling more intensely with people's search for meaning and hope in a precarious Japan. The RCJ must put its dogmatics and faith declarations even more into concrete action. The RCJ is gospel-centered and appreciates the contribution of other evangelical churches and their approach to diaconal activities in the Japanese context. Many such evangelically orientated churches base their diaconal involvements on recent declarations such as *the Cape Town Commitment* of the Lausanne Movement (which was translated into Japanese at an early stage), albeit not always overtly so. The current RCJ Moderator, Prof. Yasuhiro Hakamata (2021) also situates church-based *diakonia* within the framework of European Reformed Churches' confessional heritage, in particular the *Barmen Confession* and the *Westminster Confession of Faith*.

The RCJ shares a history of mutual diaconal involvement of more than two decades with the Dutch Reformed Family of Churches in South Africa. From a South African perspective, for instance, the *Belhar Confession* and its emphasis on justice, reconciliation, and unity remain potently valid and relevant as a check-and-balance for the church's diaconal practices. The author has actively served on the RCJ Synod Diaconal Activities Committee for the past seven years and acts as a bridge-builder between the Japanese and South African contexts in this regard.

*4.3. RCJ Case Study: Seikeikai Seeking Justice and Dignity for Vulnerable People*

Seikeikai Social Welfare Institute—previously known as Sekeikai Vocational Aid Centre, hereafter referred to as Seikeikai—was started by Rev. Makio Ihara (1926–1994), who suffered from muscular dystrophy that developed when he was 16 years old. Ihara became an RCJ pastor after graduating from Kobe Reformed Theological Seminary, overcoming many challenges as a person living with a disability. He began working at RCJ Tadanoumi congregation in 1951. Seven years later he started typewriting with his wife at the church, to earn money for their daughter's tuition.

At that time, people with disabilities were generally hidden from the public in Japan. Most people living with disabilities were unable to receive an education until much later (1979), when schools for people with disabilities became mandatory. Many people with disabilities were not actively taken out in public by their families, and often neighbors did not even know of their existence. They were not allowed to get married or have children. Fundamentally, their human dignity was denied. This lack of opportunity to find meaningful work, combined with Ihara's timely typewriting endeavor that was earning a good income, attracted the attention of people with disabilities in that area and eventually led to Seikeikai's founding notion: "If there is no place to work, let's make one ourselves" (Yoshida 2016, p. 24) (my translation).

The National Pension Law came into effect in 1961, and some people living with disabilities intentionally did not work, relying instead on disability pension. However, they wanted to live as active members of society, even if it meant doing basic, menial tasks. Seikeikai was officially established in 1960 as a facility to support the independence of people with disabilities by helping them acquire skills (in the town of Tadanoumi, near Hiroshima). The training program at Seikeikai consisted of three pillars: functional training, vocational aid, and biblical guidance (daily worship). Training began with reading and writing skills, and trainees' attitudes gradually shifted through the program's three pillars. The slogan of Seikeikai changed from "I don't want to be a cripple" to "I want to be a taxpayer". That was a significant change, which Ihara ascribed to the daily Bible teachings that positively influenced trainees' attitudes (Seikeikai 2015, p. 62). Missionaries from the Presbyterian Church (USA) and Christian Reformed Church (North America) also made constructive contributions to functional training.

In an empirical study of Rev. Ihara's work, RCJ pastor Hiromu Yamaguchi (2022) focuses on the theology behind his diaconal practices at Seikeikai—as performed by Ihara himself and under his influence—to provide insights into the characteristics of *diakonia* based on Reformed theology. Yamaguchi explored Ihara's theology by analyzing his lectures, notes, as well as his (posthumous) book, published at Seikeikai on its 55th anniversary. Furthermore, Yamaguchi conducted 15 (semi-structured and unstructured) interviews with Ihara's family members, former trainees, and employees that worked alongside him at Seikeikai.

Ihara has indeed opened the way for many people living with disabilities to become independent, go out into the world, and even change their local communities. Yamaguchi (2022, pp. 32–33) highlights two distinctive attitudes of Ihara that were ever-present. One was Ihara's view of God and people. The other was his unwavering principle-based convictions. His work was based on God's sovereignty and the gracious election of his people, and his theology was guided by the existential question, "Why and for what purpose do people with disabilities exist?" Ihara accepted his own disability, after a long struggle to

find the answer to this question, through the grace of God's election. He saw the God-given purpose and *raison d'etre* of people with disabilities in John 9:3: "Neither this man nor his parents sinned," said Jesus, "but this happened so that the works of God might be displayed in him". Ihara (2015, p. 14) phrases the matter as follows:

> It is true that life remains a mystery to us, that God's acts cannot be measured by our small scale, therefore we cannot fully comprehend His will. But we want to leave everything in God's hands and entrust our lives to His will. Since our God is the Triune God, it is only natural that He should guide our lives in a way that we cannot grasp, because everything in our lives is included in His plan. (my translation)

Furthermore, Ihara strongly believed that God loves us as worthy human beings, created with a purpose in His likeness. He believed God is a God of justice who opposes those who discriminate according to earthly standards. In line with this conviction, Ihara (quoted in Yamaguchi 2022, p. 42) asserted:

> When we think of human rights, we always presume the "image of God". We are to be respected because we bear the image of God. We must correct our views and attitudes toward all people with disabilities. I hesitate to be so explicit, but I am a person living with disability myself, so I often say, "This body (of mine) may be distorted, but the image of God that resides in it, is not broken down." We would do well to remember this. (my translation)

Ihara's work, seen from the perspective of *diakonia*, was not merely the work of philanthropic assistance and care. He not only preached about diaconal practices, but he also embodied them and served as an advocate on behalf of many vulnerable fellow brothers and sisters. Ihara viewed people with disabilities as prophets who testify to God's revealed truths with their own bodies. Moreover, Ihara believed that a society in which people with disabilities coexist with able-bodied people as dignified human beings is the ideal society expressing the will of God (Ihara 2015, p. 28).

In Ihara's lecture transcripts and writings, the concept of *diakonia* is rarely directly mentioned. However, Ihara's theology and practice were based on the constructive acceptance of his disability and the embodiment of the characteristics of Biblical *diakonia*. Ihara divided the work of the church into four categories: worship (*leitourgia*), witness (*marturia*), fellowship (*koinonia*), and service (*diakonia*). He further asserted that these four dimensions are fulfilled when they are performed in a proper balance (Seikeikai 2015, p. 351). This intricate relation between these four dimensions is affirmed widely today by those who study and practice *diakonia* intentionally (see, for instance, Nordstokke 2013, pp. 287, 297).

The *diakonia* based on Reformed theology that was demonstrated through Ihara and Seikeikai was/is not merely loving service for the sake of individuals. It is a *diakonia* with the broader societal transformation—towards justice and dignity—in its scope, toward the perfection of the Kingdom of God through Christ. Seikeikai trainees, who started to walk as human beings with dignity, began to transcend the barriers that separated them from the community. In recent years the annual Seikeikai Cultural Festival has at times (before the Corona pandemic) attracted more than 1000 people from the community, having become a beacon of hope that has broken down barriers of societal discrimination against people with disabilities. This broader effect echoes the conviction of Stanard (2015, p. 8) that, "Diakonia, therefore, is not an end in itself, but rather an instrument used by God, together with others, to build an inclusive and just community, an *oikos*, a household in which the entire creation is included, enjoying the fullness of life intended for all".

In today's complex society, welfare services are also required to have a higher level of expertise. With the support of the local community, Seikeikai has gradually transformed into a large welfare institute with many diverse projects (see https://seikeikai.ecweb.jp, accessed on 12 March 2023). Although today only a handful of its staff are overtly Christian, they are called to Christian Social Welfare work and thereby participate in the building of God's Kingdom for His glory in their community. The RCJ today continues its close rela-

tionship with Seikeikai. Many of its members support Seikeikai's work financially through church offerings, participate in its activities as volunteers, promoting its diaconal involvement in the region and beyond.

*4.4. Context Matters: Diakonia in Japan Compared to Western European or Other Settings*

The *Inner Mission* under the influence of Johan Hinrich Wichern (1808–81) was the key diaconal actor in Germany during the mid to late nineteenth century. According to Nordstokke (2020, p. 175)

> Diaconal institutions therefore grew and developed independently from official church structures. As public welfare services began to be introduced in some of the North European countries by the end of the 19th century, it became *natural* for the diaconal institutions to cooperate with them, and also for diaconal workers to find work there, with the consequence that in some cases more conections were developed between diakonia and state than between diakonia and the official church. The diaconal movement became a leading force in developed professional health and social services in Europe in the 19th century. (my italics)

Discourses about *diakonia* have progressed significantly (over almost two centuries) in Germany. Diaconal services have become an unmistakable, "natural" part of the wider European church-state-community situation. But the same is not true in Japan, where the formal connections between public welfare services and churches only started being considered after WWII. Those fragile connections are still being forged.

In multiple global contexts, churches are called to create networks in the nexus between their diaconal practices and local communities. Christians embody there the love and compassion of Christ in so-called "social spaces". Johannes Eurich (2020), current director of the University of Heidelberg's Institute for Diaconal Studies (DWI) uses this concept (of social spaces) fruitfully when he analyses it in Western European contexts regarding the collaboration between local churches and diaconal organizations. In those contexts, many churches have outsourced their specialized diaconal care to diaconal agencies that have become "professionalized service providers that operate according to the logics of social markets and professional standards. Their connection to the church has been reduced to institutional links, so that there is a discussion about the diaconal profile of such market-oriented diaconal organizations" (Eurich 2020, p. 2).

Considering the above-mentioned in connection with Japan: social spaces in Japan are not as accessible as for instance in many European (post-Christendom) countries, where Christian volunteerism manifests easier because of a long history of well-established professional diaconal agencies. Although the natural connection between churches and local communities/society—including the creation of new social spaces—are facing fresh contemporary challenges in European contexts as well, those frayed connections do not approach equal comparison with the huge gaps that exist between (or even the total lack of) state-church-community relations in Japan.

Christian Social Welfare organizations face huge, unique challenges in a Japanese context where Christians represent a radical minority. Churches are socially marginalized and do not wield much socio-political power in society, most are indeed fighting for survival. It is not easy to leverage enough resources—primarily in terms of human capital (trained staff)—to create influential networks in communities. Collaborative partnerships are not fostered easily. Notwithstanding, churches are called afresh to respond in the best ways possible and to seek and create opportunities to serve in a Japanese context fraught with uncertainties, disconnectedness, and existential loneliness (as was made clear in the first part of the article).

However, it takes a significant amount of time to foster connections and create meaningful social space networks between churches and public/social welfare organizations in Japan. A church such as the Reformed Church in Japan is one case in point. Christians in the RCJ are re-examining the theological bases on which they had been/have not been actualizing social work. In the declaration of the 20th anniversary of the RCJ's foundation

(in 1966), the notion of loving deeds (*ai no waza*) and Christian witness (evangelization) was already presented as two sides of the same (gospel-)coin, as follows:

> Concerning the practice of Christian evangelism, the teaching and example of the Lord Jesus Christ shows [us] that it should not only happen through the Word, but also by acts of love. The evangelism of our Church must also be a unified practice of theology and diaconal service of love. (RCJ 2016, my translation)

This declaration was written during the 1960s, when the RCJ's awareness of the importance and need for diaconal work started. Missionaries had created the seedbed and systems—through Christian Social Welfare agencies—for such an awareness to grow (as was explained earlier). However, within the RCJ during the first postwar decades Christian Social Welfare initiatives were not yet directly connected to the church and properly understood within the explicit framework of *diakonia*. Some of the pioneers of the RCJ—like Rev. Makio Ihara (Seikeikai) and Rev. Terunori Aoyama who founded the RCJ Shizuoka Centre for the Blind—were attuned to the needs of vulnerable people in Japanese society. They and their groundbreaking work were/are a gift from God to the RCJ.

Eventually, various natural disasters (especially since 1995) have let the "wave of awareness" break onto the "shore", i.e., onto churches' doorsteps. These crises forced open a fresh realization of the potential and significance of every Christian believer's diaconal responsibility and calling, as opposed to some high ideals that only a few selected individuals can attain. The RCJ (among other churches in Japan) is realizing anew that *diakonia* belongs to the core of the Church's existence. Evangelism today is viewed differently than in the immediate postwar "Christian boom" era when evangelistic mega-events were the order of the day. Now, instead, small evangelistic churches realize the need to be open to the immediate community, e.g., by building bridges through kindergarten cafeterias our soup kitchens. All such congregations will not (yet) necessarily call such ministries *diakonia*, but they are essentially equal to diaconal work, sometimes as part of Christian Social Welfare, other times in partnership with secular NGOs and so forth.[1]

In South Africa, the Dutch Reformed Church is also making a deliberate turn to re-envision *diakonia* and create new networks for its missional-diaconal practices in social spaces (see Van der Watt 2019). In other parts of the world (e.g., in the UK) the same activities might be called integral or holistic missions or even the social responsibility/ministries of the church (e.g., in the US). New discourses about *diakonia* abound and this is indeed a global, evolving conversation.

## 5. Conclusions

This article gives a descriptive overview of post-WWII developments and the current state of church-related diaconal initiatives in Japan. *Diakonia* is indeed at the heart of the Church but simultaneously reaches far beyond the Church's institutional frameworks. The central research question has been: why is there a complicated relationship between church-based practices defined as *diakonia*, and the work of Christian-based social welfare organizations, in the post-WWII Japanese context?

The basic finding is that such a problematic relationship exists primarily because there are indeed many differences between the practices and approaches of, on the one hand, pioneering individuals through Christian Social Welfare, and, on the other hand, congregations (this article focused on the RCJ) that were/are involved in diaconal activities in postwar Japan. Although the theological foundations of church-based *diakonia* and Christian Social Welfare agencies share a common, ecumenically validated origin (i.e., the Bible and the example of Christ, the True *Diakonos*), these respective expressions of the *Missio Dei* show disparities.

It was further assessed that a tension field remains between the state and church in terms of their respective expectations and foundational principles. In Japan (different from many European countries) Christian volunteerism does not occur readily because of a long history of well-established professional diaconal agencies. Correspondingly, sizeable gaps characterize state-church-community relations in Japan. Moreover, because churches are

mostly socially marginalized and do not possess much socio-political power in society, it is deeply problematic to raise enough resources to create influential networks in communities. Collaborative partnerships are not fostered easily.

Nevertheless, this study points out that there is a need to expose Christians to diaconal care experiences as volunteers, individually or as organizations. In recent years the concept *diakonia*—enacted through a variety of diaconal practices—has become more widely used in the RCJ. Diaconal involvements via the RCJ's connections in South Africa have existed for more than two decades. Recent RCJ activities include diaconal study tours to, and volunteer work in, South Africa and Cambodia as well as financial support to Myanmar and Turkey/Syria. As a result, there is a new awareness, albeit a gradual awakening, of *diakonia* as a core element of being Church, in local communities and beyond.

Sharing the gospel through diaconal activities exceeds, but does not necessarily exclude or hinder, the affirmation of a spoken message or the sharing of God's Word. Amid the pain of social isolation and loneliness, abandonment and hopelessness experienced by many in Japan, the following questions are posed: how can the church be(come) a place of meaningful connectedness (*ibasho*), where lonely people gain social recognition, belonging, and hopeful support in challenging times? How can Christians be more intimately involved in the existential realities in society and grapple more intently with people's search for meaning and hope in a precarious Japan, where hope for anything beyond private acquisition is apparently fading away for so many?

This article also examined the influence of some leading Christian Social and Public Welfare practitioners and scholars. Collaterally, it was concluded that the Christian motivation behind welfare practices—nowadays increasingly defined under the rubric of *diakonia*—often serves as the root of volunteerism. The claim followed that *diakonia* should primarily be understood as church-based activities of care, especially within the fields of education, social work, and physical/mental health. Such activities can be performed by all Christians (lay and ordained), both collectively as local congregations/faith communities, as well as by individuals through professional social welfare agencies, if/where such agencies exist.

In the last part of the article, a case study from within the Reformed Church in Japan was examined. Seikeikai Social Welfare Institution's work and the theological inspiration and depth of its founder, Rev. Makio Ihara, were presented to highlight the quest for justice and human dignity, specifically for vulnerable people. The initiatives of Ihara and Seikeikai, viewed from the perspective of *diakonia*, were found to be more than philanthropic assistance and care for the sake of individuals. Ihara embodied diaconal practices and served as a prophetic advocate on behalf of many vulnerable people living with disabilities.

The ongoing challenge in Japan, of finding a constructive balance between serving Christ/Church and serving the Emperor/State, remains. The article illustrates that Christian individuals and Christian-based welfare organizations are not merely serving the Japanese state. Instead, such individuals and/or organizations served/are serving people in need because of their biblically founded convictions (to serve their neighbors)—sometimes explicitly defined as *diakonia*, but often not—within the restrictions and with the support of government resources.

The important task of theological reflection on practices of *diakonia* remains, namely on who God is, who we are as humans, what situations vulnerability and need exist, and how we can care justly in the midst thereof. Christians need to constantly examine the theological bases on which they conduct social work, as well as the methods they use to do so. Diaconal churches are called to creatively and constructively connect social welfare and Christianity as a serving community that embodies the love and compassion of Christ, whilst keeping a critical distance towards societal systems and structures. This article makes a small contribution to this ongoing task in an interdisciplinary way within the Japanese context.

**Funding:** This research received no external funding.

**Acknowledgments:** I gratefully acknowledge the technical support (through copy editing) of J. Nelson Jennings.

**Conflicts of Interest:** The author declares no conflict of interest.

## Note

[1] These insights were gleaned from a semi-structured interview with a previous moderator of the RCJ (currently the dean of Kobe Reformed Theological Seminary), Prof. Takashi Yoshida about this denomination's historical cognizance of/involvement in diaconal activities. Prof. Yoshida is also the founder of *Tohoku Help*, an ecumenical Christian aid organization that is still—12 years after 3.11—involved with supporting the North-Eastern Japan disaster victims.

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
