# Peer review of "Diaconal Church Initiatives and Social/Public Welfare in Postwar Japan: A Descriptive Overview"

_religions, doi:10.3390/rel14050594_

Round 1

Reviewer 1 Report

Diaconal Church initiatives

The article investigates the relationship between the work of Christian based social welfare organizations and church base practises defined as diakonia in Japan. The research question is: Why is the relationship complicated?

To answer this question the author in the first place looks at the sosio-economic realities after WWII and what it means for the care of people in need. Then the author describes the post war initiatives of Christian welfare organisations in the light of the complexity of religion in Japan and Christians as a small minority. In the last place the author investigates the church based diaconal responses in Japan.

Based on the information gleaned from these four investigation areas the author makes a comparison between the Christian welfare organisations and the church based diaconal responses to answer the research question.

The article is logically structured and easy to read. The function of each of these research areas with the sub-areas are clear leading up to the conclusion. The use of relevant sources is scientifically acceptable and enough. The author has consulted the latest sources and he/she stayed focussed on answering the research question but also gives an indication that there are related research areas for which sources are recommended.

The reason why Christian welfare organizations developed more toward greater contact with the state than with the churches are describe satisfactorily.

Things to consider

The author concluded that “The fundamental dilemma in Japan, of finding a constructive balance between serving Christ/Church and serving the Emperor/State remains.”

The question is if this is truly the “fundamental dilemma” in the relationship between the Christian welfare organizations and the church based diaconal work. Are the Christian welfare organizations only serving the state or are they serving people in need on the bases of the Bible only within the limits and means provided by the state.

It will be important for the quality of the article that the author reconsider the whole conclusion.

The conclusion does not really answer the research question and does not contribute to a solution.

It is as if the author lost perspective on the rich data that were given in the article. The author should read his/her own article again and rewrite the conclusion focussed on answering the research question considering the wealth of data in the previous sections.

Line 380 should be completed.

Conclusion

A well written article that only needs to attend to these recommendations

Author Response

Dear Reviewer 1,

  1. Thank you for your constructive review and helpful comments.
  2. As suggested, I reconsidered the whole conclusion so that it gives a more applicable answer to the main research question.
  3. The article was intentionally descriptive, not prescriptive, therefore I did not focus on suggesting concrete answers or clear solutions to the complex issues presented throughout.
  4. I completed line 380, as suggested.

Reviewer 2 Report

The article has some minor problems with English grammar and syntax throughout, suggesting that it is not the product of a native-speaker or that it was not carefully proofed by the author. For example, in an important opening statement, the article reads,

In this article diakonia understood as a call to action – arising from the gospel of Jesus Christ – which incorporates care for 46 God’s whole creation and seeks to combat many forms of injustice and vulnerability. 

Missing in the above sentence is the verb "is" which is needed prior to "understood" or the sentence does not make sense. Readers can figure out what is being said, provided they are native speakers, but will think that the article has not been sufficiently proofread and revised. There are other such minor but important issues with the article. Still, I have a very high opinion of it.

The article uses the past tense on occasion when I think the present tense would be better. For example, the opening sentence of the abstract reads,

This article reflected on post-WWII developments and the current state of church-related diaconal initiatives in Japan. 

The present tense "reflects" would be better than the past tense "reflected." I strongly suggest changing this. Minor point but important. 

And, the last sentence of the article reads:

This article made a small contribution to this ongoing task in an interdisciplinary way within the Japanese context. 

I think "makes" (present tense) would be better than "made" (past tense) here. This is a minor point but changing the tense would improve the article somewhat. 

The references need to be formatted with hanging paragraphs rather than indented ones. The hanging paragraph makes the author's identity (or the entry lead's identity) more apparent. 

My comments here are the same as above: 

The article has some minor problems with English grammar and syntax throughout, suggesting that it is not the product of a native-speaker or that it was not carefully proofed by the author. For example, in an important opening statement, the article reads,

In this article diakonia understood as a call to action – arising from the gospel of Jesus Christ – which incorporates care for 46 God’s whole creation and seeks to combat many forms of injustice and vulnerability. 

Missing in the above sentence is the verb "is" which is needed prior to "understood" or the sentence does not make sense. Readers can figure out what is being said, provided they are native speakers, but will think that the article has not been sufficiently proofread and revised. There are other such minor but important issues with the article. Still, I have a very high opinion of it.

The article uses the past tense on occasion when I think the present tense would be better. For example, the opening sentence of the abstract reads,

This article reflected on post-WWII developments and the current state of church-related diaconal initiatives in Japan. 

The present tense "reflects" would be better than the past tense "reflected." I strongly suggest changing this. Minor point but important. 

And, the last sentence of the article reads:

This article made a small contribution to this ongoing task in an interdisciplinary way within the Japanese context. 

I think "makes" (present tense) would be better than "made" (past tense) here. This is a minor point but changing the tense would improve the article somewhat. 

The references need to be formatted with hanging paragraphs rather than indented ones. The hanging paragraph makes the author's identity (or the entry lead's identity) more apparent. 

Author Response

Dear Reviewer 2,

  1. Thank you for your review and feedback.
  2. As suggested, I thoroughly addressed the problems concerning grammar and syntax.
  3. I formatted the references with hanging paragraphs.

Reviewer 3 Report

I have reviewed this article about the modern evolution of Christian

Church in Japan. The topic has been thoroughly studied in the

article. This manuscript is a historical review article that I find knowledgeable and informational. In addition, the article's methodology is well-focused on examples that justify how the current situation has been reached.

I found the methodology correct, and also the discussion of the cases.

The article, in summary, seems adjoined and authoritative; only I

would suggest tracing more relations and joint studies with other

types of “churches” of modern Japan.

The conclusions are consistent, and so are the results.

Summary of evaluation: This article is indeed interesting. It could be

published after MINOR adjustments.

Author Response

Dear Reviewer 3,

  1. Thank you for your review and positive feedback.
  2. Unfortunately, I was not able to trace more relations and joint studies with other types of churches of modern Japan.